NGScloud2: optimized bioinformatic analysis using Amazon Web Services

Mora-Márquez Fernando 1
Vázquez-Poletti José Luis 2
López de Heredia Unai unai.lopezdeheredia@upm.es 1
1 GI Sistemas Naturales e Historia Forestal, Dpto. Sistemas y Recursos Naturales, ETSI Montes, Forestal y del Medio Natural, Universidad Politécnica de Madrid , Madrid , Spain
2 GI Arquitectura de Sistemas Distribuidos, Dpto. de Arquitectura de Ordenadores y Automática, Facultad de Informática, Universidad Complutense de Madrid , Madrid , Spain
Orlov Yuriy
Electronic publication date: 2021 Apr 16
Publication date: 2021
Volume: 9
Electronic Location ID: e11237
Received 2020 Dec 8; Accepted 2021 Mar 17
Copyright: ©2021 Mora-Márquez et al.
Copyright year: 2021
Copyright holder: Mora-Márquez et al.
License: This is an open access article distributed under the terms of the Creative Commons Attribution License, which permits unrestricted use, distribution, reproduction and adaptation in any medium and for any purpose provided that it is properly attributed. For attribution, the original author(s), title, publication source (PeerJ) and either DOI or URL of the article must be cited.
License URL: https://creativecommons.org/licenses/by/4.0/

Keywords: AWS, Bioinformatics, Cloud computing, Functional annotation, Next generation sequencing, Transcriptomics

Funding: Spanish Ministry of Economy and Competitiveness-MINECO AGL2015-67495-C2-2-R Spanish Ministry of Science and Innovation PID2019-110330GB-C22 RTI2018-096465-B-I00 Regional Government of Madrid P2018/TCS4499 S2018/TCS-4499 Amazon Research Grant This work was supported by the Spanish Ministry of Economy and Competitiveness-MINECO [grant number AGL2015-67495-C2-2-R]; the Spanish Ministry of Science and Innovation [grant numbers PID2019-110330GB-C22 and RTI2018-096465-B-I00]; the Regional Government of Madrid [grant numbers P2018/TCS4499 and S2018/TCS-4499]; and by an Amazon Research Grant. The funders had no role in study design, data collection and analysis, decision to publish, or preparation of the manuscript.

==============================
Background

NGScloud was a bioinformatic system developed to perform de novo RNAseq analysis of non-model species by exploiting the cloud computing capabilities of Amazon Web Services. The rapid changes undergone in the way this cloud computing service operates, along with the continuous release of novel bioinformatic applications to analyze next generation sequencing data, have made the software obsolete. NGScloud2 is an enhanced and expanded version of NGScloud that permits the access to ad hoc cloud computing infrastructure, scaled according to the complexity of each experiment.

Methods

NGScloud2 presents major technical improvements, such as the possibility of running spot instances and the most updated AWS instances types, that can lead to significant cost savings. As compared to its initial implementation, this improved version updates and includes common applications for de novo RNAseq analysis, and incorporates tools to operate workflows of bioinformatic analysis of reference-based RNAseq, RADseq and functional annotation. NGScloud2 optimizes the access to Amazon’s large computing infrastructures to easily run popular bioinformatic software applications, otherwise inaccessible to non-specialized users lacking suitable hardware infrastructures.

Results

The correct performance of the pipelines for de novo RNAseq, reference-based RNAseq, RADseq and functional annotation was tested with real experimental data, providing workflow performance estimates and tips to make optimal use of NGScloud2. Further, we provide a qualitative comparison of NGScloud2 vs. the Galaxy framework. NGScloud2 code, instructions for software installation and use are available at https://github.com/GGFHF/NGScloud2. NGScloud2 includes a companion package, NGShelper that contains Python utilities to post-process the output of the pipelines for downstream analysis at https://github.com/GGFHF/NGShelper.

Introduction

Next Generation Sequencing (NGS) has largely allowed the development of genomics and transcriptomics, and many experiments based on this methodology are routinely performed in many fields of biological and life sciences (Frese, Katus & Meder, 2013). The way these experiments are conducted is not, however, trivial, and requires the use of suitable methods at all stages of the experiment to produce sound results. Moreover, NGS experiments must be dimensioned properly to be cost-efficient (Wordsworth et al., 2018).

A generic NGS experiment can be divided into two phases (López de Heredia, 2016). First, an in vitro phase that consists of the construction of genomic libraries (sets of nucleic acid fragments processed in the laboratory following the methodological instructions required by the sequencing technique used in the experiment) and sequencing of those libraries in an NGS platform to generate read files (see Heather & Chain, 2016 for a description of the available sequencing platforms). Then, an in silico or bioinformatics phase is needed to process read sequencing files output by sequencing platforms. The bioinformatic analysis can be subdivided into three stages:(1) pre-processing of read files; (2) read assembly or read mapping; and (3) post-processing of assembly/mapping results. In the pre-processing, the quality of the raw reads (those generated by the NGS platform without any modification) is evaluated. The causes that may produce a decrease in read quality and introduce bias in further inference, such as the presence of adapters and other sequences used in the construction of the libraries, bases with poor quality, PCR duplicates and possible experimental artifacts, etc., are eliminated. When there is not a reference genome or transcriptome, the pre-processed reads are assembled into larger fragments, or clustered, reconstructing the original DNA chains, originating contigs (continuous sequences obtained from the superposition of multiple reads) and scaffolds (ordered distribution of contigs that is inferred when paired reads are used maintaining gaps between them). When the reference genome/transcriptome exists, the pre-processed reads are mapped to the genome to determine the exact genomic region where they align. The post-processing stage will depend on the specific methodology employed in the analysis and on the aims of the experiments. Frequently, this stage consists of the assessment of the quality of the assembly/mapping, on the application of subsequent filters to find significant genomic variants or changes in expressional patterns, and on a functional annotation step to determine the biochemical and biological function of the post-processed sequences.

The large output size of NGS technologies and the algorithms and applications employed in their analysis, present processing limitations typical of big data, such as RAM size, CPU capacity, storage and data accessibility (Yang, Troup & Ho, 2017). Therefore, research labs have to allocate a significant part of their budget to provisioning, managing and maintaining their computational infrastructure (Kwon et al., 2015).

A cost-efficient alternative for bioinformatic analysis of NGS data that presents several advantages over local or HPC hardware infrastructure resides in cloud computing (Langmead & Nellore, 2018), and applications and platforms are optimized to operate bioinformatics on cloud systems, such as the Galaxy framework (Afgan et al., 2018). Cloud computing is flexible and scalable, allowing various configurations of OS, RAM size, CPU number and almost unlimited storage to fit the hardware resources for a specific bioinformatics workflow. Once the workflow computing requirements are provisioned, hardware resources are readily available, and the workflow performance and data can be securely accessed and monitored at any time from any local computer with internet access. Moreover, for public cloud services, the user only pays for the effectively used resources, reducing experiment times and costs.

Amazon Web Services (AWS) is a public cloud computing platform that has a large number of information technology infrastructure services. The Elastic Compute Cloud (EC2) is the AWS service that provides computing capacity adjustable to the NGS experiment requirements with a wide range of instances, currently more than 350 models with specific processors, number of CPUs, RAM quantity and network type (https://aws.amazon.com/ec2/). EC2 instances (virtual machines) can be grouped into: (1) general purpose instances (types t2, t3, t3a, m3, m4, m5 and m5a, among others), with a balanced mix of hardware resources; (2) compute optimized instances (types c3, c4, c5, and c5a, among others), with high-performance processors and a higher ratio of CPUs per RAM memory than instances for general use; and (3) memory optimized instances (types r3, r4, r5 and r5a, among others), indicated for workloads with high volume of data and with higher ratio of RAM memory per number of CPUs than general purpose instances. A general-purpose instance typically has 4 GiB of RAM per CPU; a compute optimized instance, 2 GiB per CPU; and a memory optimized instance, 8 GiB per CPU. In addition to the previous groups, there are accelerated computation instances that have hardware accelerators or co-processors and optimized storage used for large volumes of data. The purchasing option of an instance can be on-demand, with a pay-per-use model that has a fixed hourly price (https://aws.amazon.com/ec2/pricing/on-demand/); or spot, when the non-used EC2 computing capacity is requested. The spot instances have large discounts compared to the on-demand hourly pricing (https://aws.amazon.com/ec2/spot/pricing) but, they have the advantage that the instance can be stopped by AWS when it is required. The Elastic Block Store (EBS) is the AWS storage service designed to store data from the EC2 service in named data volumes (https://aws.amazon.com/ebs/), and has a fixed monthly quote per GiB.

Here we present NGScloud2, a new version of the NGScloud software (Mora-Márquez, Vázquez-Poletti & López de Heredia, 2018). NGScloud was developed as a bioinformatics system to perform de novo RNA sequencing (RNAseq) analysis of non-model species. This was accomplished using the cloud computing infrastructure from AWS (EC2) and its high-performance block storage service (EBS). NGScloud allowed to create one or more EC2 instances (virtual machines) of m3, c3 or r3 instance types forming clusters where analytic processes were run using StarCluster, an open source cluster-computing toolkit for EC2 (http://star.mit.edu/cluster/). However, NGScloud did not support the new instance types that AWS has made available since the original application release. Below we describe the major new features of NGScloud2 that significantly expand NGScloud2 functionality with respect to the original version, providing workflows for reference-based RNAseq, Restriction site Associated DNA sequencing (RADseq) and functional annotation.

Figure 1 Technical improvements of NGScloud2.

Hardware infrastructure in AWS’s cloud can be setup using a cluster mode with only the previous generation on-demand instances or a native mode that allows configuring new generation on-demand/spot instances. Apps, reads, datasets, references and results can be stored using multiple volumes or a single storage volume containing specific working directories. All the cloud configuration processes are controlled from the NGScloud2 GUI installed on a local computer.

Materials & Methods

NGScloud2 is a free and open source program written in Python3. Source code and a complete manual with installation instructions and tutorials to exploit all the potential of NGScloud2 are available from the GitHub repository (https://github.com/GGFHF/NGScloud2). NGScloud2 presents remarkable differences with respect to NGScloud both in the way AWS resources are managed to better exploit all the potential of EC2 and EBS, but also by incorporating the possibility of running a more complete set of bioinformatics applications and pipelines for de novo RNAseq, reference-based RNAseq, RADseq and functional annotation. In addition, a toolkit of Python programs useful to post-process the output of RNAseq and RADseq experiments is available in NGShelper (https://github.com/GGFHF/NGShelper).

Technical improvements

NGScloud2 introduces a more efficient architecture of instances and volumes than the original version (Fig. 1). While NGScloud used one volume for each type of existing datasets (applications, databases, references, reads and results), NGScloud2 offers the possibility of holding all dataset types in a unique volume, thus reducing the complexity in volume management. NGSCloud2 philosophy is based on the “cluster” concept. A cluster is a set of 1 to n virtual machines with the same instance type. Each instance type has its hardware features: processor type, CPU number, memory amount, etc. (https://aws.amazon.com/ec2/instance-types/).

NGScloud2 includes two cluster modes, StarCluster and native. The StarCluster mode uses StarCluster (http://star.mit.edu/cluster/), an open source cluster-computing toolkit for EC2, which implements clusters of up to 20 virtual machines, enabling faster analysis. The last version of StarCluster (0.95.6) dates from 2013 and can only use AWS’s previous generation instance types, i.e., m3, c3 or r3. In NGScloud2, we provide a patch to enable using m4, c4 and r4 instance types in the StarCluster mode.

To reduce the dependency of NGScloud from StarCluster, which only allows to create clusters of previous generation instances, NGScloud2 has incorporated a “native” instance creation mode that sets a single virtual machine with any of the currently available on-demand EC2 instance types (m4, c4, r4, m5, m5a, c5, c5a, r5 and r5a). The new generation instance types are slightly cheaper and their hardware improves over equivalent hardware from previous generations. Moreover, the new version enables launching “spot instances” that derive from unused EC2 capacity in the AWS cloud (https://aws.amazon.com/ec2/spot/). Spot instances have the advantage of being up to 50–80% cheaper than on-demand instances at the cost of suffering unpredictable interruption out of control of the user (see the main characteristics of spot instances at https://aws.amazon.com/ec2/spot/pricing/). Therefore, using spot instances is highly recommended for data transfer and for certain bioinformatics processes that run fast, process small volume input or include the possibility to be re-launched from the process interruption point.

NGScloud2 includes a user-friendly graphical front-end to operate the hardware resources, submit processes, and manage the data. The front-end includes a drop-down menu to configure AWS resources (clusters, nodes and volumes) and to install available bioinformatics software. Data transfer between the cloud and the local computer is operated through another drop-down menu. Additional menus are available to run de novo RNAseq, reference-based RNAseq, RADseq and functional annotation workflows, respectively. Log files of each executed process can be consulted in the “Logs” menu.

New methods and applications available

The other major improvements of NGScloud2 over NGScloud are related to the implementation of new bioinformatics pipelines and application tools (Table 1) that are automatically installed using Bioconda (Grüning et al., 2018), thus giving access to updated versions of the software without worrying about dependencies and software requirements. While the original purpose of NGScloud was to help in de novo RNAseq analysis, NGScloud2 includes pipelines and applications to perform reference-based RNAseq, RADseq and functional annotation. The implemented bioinformatics applications were selected through a search of the total number of citations in JCR (https://www.scimagojr.com) publications in the Web of Science. Some other utilities are original software applications implemented by our research group.

Table 1 Software applications selected for de novo RNAseq (dnRNAseq), reference-based RNA-seq (rbRNAseq), RADseq (RADseq) and taxonomy-oriented annotation (TOA) workflows in NGScloud2.

Software	Workflows	Task	Reference	# citations †	
BCFtools	dnRNAseq, rbRNAseq & RADseq	Variant calling	Danecek & McCarthy (2017)	46	
BEDtools	dnRNAseq, rbRNAseq & RADseq	Variant calling	Quinlan & Hall (2010)	7,531	
Bowtie2	dnRNAseq, rbRNAseq & RADseq	Read alignment	Langmead & Salzberg (2012)	16,309	
BLAST+	TOA	Annotation pipeline	Camacho et al. (2009)	5,964	
BUSCO	dnRNAseq	Transcript quality	Waterhouse et al. (2018)	584	
CD-HIT:CD-HIT-EST	dnRNAseq	Filtering	Li & Godzik (2006)	4,516	
Cufflinks	rbRNAseq	Cufflinks-Cuffmerge: assembly
Cuffquant: quantitation
Cuffdiff: differential expression	Trapnell et al. (2012)	6,679	
cutadapt	dnRNAseq, rbRNAseq & RADseq	Preprocessisng	Martin (2011)	23	
DETONATE:RSEM-EVAL	dnRNAseq	Assembly quality	Li et al. (2014)	125	
ddRADseqTools	RADseq	Experimental design	Mora-Márquez et al. (2017)	11	
DIAMOND	TOA	Annotation pipeline	Buchfink, Xie & Huson (2015)	2,083	
eXpress	dnRNAseq	Quantitation	Roberts & Pachter (2013)	482	
FastQC	dnRNAseq, rbRNAseq & RADseq	Preprocessisng	Andrews (2010)	127	
GMAP-GSNAP	GMAP: rbRNAseq
GSNAP: rbRNAseq & RADseq	GMAP: Transcriptome alignment
GSNAP: read alignment	Wu et al. (2016)	119	
HISAT2	rbRNAseq	Read alignment	Kim et al. (2019)	288	
HTSeq:ht-seq-count	rbRNAseq	Quantitation	Anders, Pyl & Huber (2015)	6,906	
ipyrad	RADseq	Full pipeline	Eaton & Overcast (2020)	30	
Kallisto	dnRNAseq	Quantitation	Bray et al. (2016)	1,883	
NGShelper	dnRNAseq	Transcript-filtering: filtering
transcriptome-blast: annotation	https://github.com/GGFHF/NGShelper	–	
QUAST	dnNRAseq	Assembly quality	Gurevich et al. (2013)	2,061	
RADdesigner	RADseq	Design	Guillardín-Calvo et al. (2019)	2	
rnaQUAST	dnNRAseq	Assembly quality	Bushmanova et al. (2016)	31	
SAMtools	dnRNAseq, rbRNAseq & RADseq	Variant calling	Li et al. (2009)	4,429	
SOAPdenovo2	RADseq	Pseudo-assembly	Luo et al. (2012)	2 558	
SOAPdenovo-Trans	dnRNAseq	Assembly	Xie et al. (2014)	449	
STAR	rbRNAseq	Read alignment	Dobin et al. (2013)	9,718	
Tabix	dnRNAseq, rbRNAseq & RADseq	Variant calling	Li (2011)	195	
TOA	TOA	all	Mora-Márquez et al. (2021)	–	
TopHat2	rbRNAseq	Read alignment	Kim et al. (2013)	6,912	
Trans-ABySS	dnRNAseq	Assembly	Robertson et al. (2010)	542	
TransDecoder	TOA	Annotation pipeline	https://github.com/TransDecoder	45	
Transrate	dnNRAseq	Assembly quality	Smith-Unna et al. (2016)	243	
Trimmomatic	dnRNAseq, rbRNAseq & RADseq	Preprocessing	Bolger, Lohse & Usadel (2014)	14,950	
Trinity	Trinity: dnRNAseq
genome-guided Trinity: rnRNAseq
insilico_read _normalization: dnRNAseq	Trinity: assembly
genome-guided Trinity: assembly
insilico_read_normalization: preprocessing	Haas et al. (2013)	3,330	

De novo RNAseq

The original software was mainly focused on de novo assembly of RNAseq libraries using either Trinity, and included pre-processing of reads with FASTQC (Andrews, 2010), Trimmomatic (Bolger, Lohse & Usadel, 2014) and three de novo RNAseq assemblers: Trinity (Haas et al., 2013), SOAPdenovo-Trans (Xie et al., 2014) and Trans-ABySS (Robertson et al., 2010). NGScloud2 de novo RNAseq workflow has been improved (Fig. 2) by including cutadapt (Martin, 2011) to perform read pre-processing, a new read alignment step with Bowtie2 (Langmead & Salzberg, 2012) to map back the reads to the assembled transcriptome and software to quantify total counts of transcripts for further differential expression analysis: eXpress (Roberts & Pachter, 2013) and Kallisto (Bray et al., 2016). Intensive processes, such as Trinity and SOAPdenovo-Trans transcriptome assemblers can now be re-launched from the point where the process interruption occurred, thus preventing unexpected malfunctioning of the cloud system or software bugs (Mora-Márquez et al., 2020). A variant calling step is also included to find SNPs or indels using SAMtools (Li et al., 2009), BEDtools (Quinlan & Hall, 2010) and BCFtools (Danecek & McCarthy, 2017).

Figure 2 de novo RNAseq workflow in NGScloud2.

The workflow has the possibility to select several applications to perform each step of the analysis: read pre-processing, de novo transcriptome assembly, transcriptome filtering and quality assessment, read mapping, quantitation, variant calling and transcript annotation. The selection of applications for each step in the workflow and the parameter configuration are controlled from the NGScloud2 GUI installed on a local computer.

Reference-based RNAseq

In the last years, an increasing number of genomic and transcriptomic resources are available for many plant and animal species. Therefore, reference-based RNAseq is expected to become a usual practice not only for model species. NGScloud2 includes a workflow to accomplish read pre-processing, read alignment, reference-guided assembly, quantitation, differential expression and variant calling (Fig. 3). Read pre-processing is done with the same tools as for de novo RNAseq (Trimmomatic and cutadapt). Read alignment to a reference genome assembly can be performed with Bowtie2, or with popular splice-aware aligners: Hisat2 (Kim et al., 2019), TopHat2 (Kim et al., 2013), STAR (Dobin et al., 2013) or GSNAP (Wu et al., 2016). Moreover, trancriptome alignments can also be run against a reference genome using GMAP (Wu et al., 2016). After read alignment, a transcriptome can be assembled using Cufflinks-Cuffmerge (Trapnell et al., 2012). Reference-guided de novo assembly can also be performed with Trinity’s genome guided version (Haas et al., 2013). Transcript or isoform abundance can be quantified with Cuffquant (Trapnell et al., 2012) or HT-seq-count (Anders, Pyl & Huber, 2015), and differential expression analysis can be run with Cuffdiff and Cuffnorm (Trapnell et al., 2012), or the expression matrix can be downloaded to run locally more up-to-date differential expression packages, such as DESeq2 (Love, Huber & Anders, 2014) or edgeR (Robinson, McCarthy & Smyth, 2010). A variant calling step that operates in a similar way than for de novo RNAseq is also included.

Figure 3 Reference-based RNAseq workflow in NGScloud2.

The workflow has the possibility to select several applications to perform each step of the analysis: read pre-processing, read alignment, transcriptome assembly, transcript alignment, variant calling quantitation and differential expression. The selection of applications for each step in the workflow and the parameter configuration are controlled from the NGScloud2 GUI installed on a local computer.

RADseq

Another major novelty in NGScloud2 is the possibility of running RADseq bioinformatics workflows. This reduced genome representation methodology and its derivates (e.g., ddRADseq) are used to find out polymorphism in specific genomic regions nearby restriction enzyme cut sites in populations of multiple individuals, and has revealed powerful in phylogenetics, population genetics, and association mapping studies, among others (Andrews et al., 2016). In NGScloud2, we have included ddRADseqTools (Mora-Márquez et al., 2017) and RADdesigner (Guillardín-Calvo et al., 2019) to assess the optimal experimental design of a RADseq experiment, i.e., to choose the enzyme combinations, simulate the effect of allele dropout and PCR duplicates on coverage, quantify genotyping errors, optimize polymorphism detection parameters or determine sequencing depth coverage.

The workflow of RADseq data in NGScloud2 allows to analyze the data using two strategies (Fig. 4). RADseq libraries can be mapped with Bowtie2, GSNAP or HISAT2 to an available genome or pseudogenome assembly. The pseudogenome can be assembled using the same (or complementary) reads with SOAPdenovo2 genomic assembler (Luo et al., 2012), or with the Starcode sequence clusterizer (Zorita, Cuscó & Filion, 2015). After read mapping, variant calling is performed in a similar way than for de novo RNAseq. The alternative is to perform read clusterization, filtering and variant calling in a single step with the robust iPyrad pipeline (Eaton & Overcast, 2020).

Figure 4 RADseq workflows in NGScloud2.

The reference-based RAD-seq workflow has the possibility to select several applications to perform each step of the analysis: RAD-seq experiment design, read pre-processing, pseudo-assembly, read alignment, and variant calling. Alternatively, NGScloud2 allows to run the full iPyrad pipeline on the cloud. The selection of applications for each step in the workflow and the parameter configuration are controlled from the NGScloud2 GUI installed on a local computer.

Functional annotation

As a last improvement over the original version, NGScloud2 encapsulates our standalone application TOA (Taxonomy-oriented annotation) (Mora-Márquez et al., 2021), so it can run in EC2. This application automates the extraction of functional information from genomic databases, both plant specific (PLAZA) and general-purpose genomic databases (NCBI RefSeq and NR/NT), and the annotation of sequences (Fig. 5). TOA is a good complement for both RNAseq and ddRADseq workflows in non-model plant species that has shown optimal performance in AWS’s EC2 cloud. TOA aims to establish workflows geared towards woody plant species that automate the extraction of information from genomic databases and the annotation of sequences. TOA uses the following databases: Dicots PLAZA 4.0, Monocots PLAZA 4.0, Gymno PLAZA 1.0, NCBI RefSeq Plant and NCBI Nucleotide Database (NT) and NCBI Non-Redundant Protein Sequence Database (NR). Although TOA was primarily designed to work with woody plant species, it can also be used in the analysis of experiments on any type of plant organism. Additionally, NCBI Gene, InterPro and Gene Ontology (http://geneontology.org/) databases are also used to complete the information.

Figure 5 Functional annotation workflow in NGScloud2.

The functional annotation workflow allows running TOA (Taxonomy-oriented annotation) in AWS’s cloud. Pipelines to run functional annotation are executed in the cloud. The databases of reference and the order they are explored can be configured. Query sequences (transcripts or DNA fragments) are aligned to TOA database using either BLAST+ or DIAMOND, and functional annotation information from several ontology systems (GO, KEGG, E.C., MetaCYC) is extracted to annotation report files that can be merged and used to build statistic files and ready-to-publish figures. The load of external genomic databases into TOA database, the selection of pipelines and the parameter configuration of TOA runs are controlled from the NGScloud2 GUI installed on a local computer.

NGShelper

Besides the cloud infrastructure deployed in NGScloud2, we have included a companion package, NGShelper that contains Python utilities to post-process the output of NGScloud2 pipelines. The package contains some Bash (Linux) and Bat (Windows) scripts to facilitate running the Python3 programs.

NGShelper facilitates format conversion of output files, filtering and subsetting of results, VCF and FASTA files statistics extraction, among others. Utilities list and their usage and parameters can be consulted at https://github.com/GGFHF/NGShelper/blob/master/Package/help.txt.

Validation of NGScloud2

The correct operability of the pipelines for de novo RNAseq, reference-based RNAseq, RADseq and functional annotation was tested with data generated by our research group. Test data for RNAseq and RADseq workflows consisted of two sets of Illumina reads: (1) Pcan, a paired-ended RNA library of xylem regeneration tissue of the conifer tree Pinus canariensis (Mora-Márquez et al., 2020). (2) Suberintro, a set of 16 paired ended Illumina libraries of Quercus suber, Quercus ilex and their hybrids obtained from leaf tissue; eight libraries correspond to genotyping-by-sequencing with MslI and other eight libraries correspond to ddRADseq with PstI-MspI (see details in Guillardín-Calvo et al., 2019). Read data are available at NCBI: SRX5228139-SRX5228161 for Pcan, and SRX5019123-SRX5019138 for Suberintro. The functional annotation workflow was tested with a small subset of transcripts corresponding to the monolignol biosynthesis gene family in Arabidopsis (Raes et al., 2003).

We further compared the capabilities of NGScloud2 as compared to other bioinformatics platforms that make use of cloud systems, such as the popular Galaxy framework (Afgan et al., 2018) or the iPlant Atmosphere service of Cyverse (Skidmore et al., 2011).

Results

Validation of NGScloud2 operability

We have checked the correct operability of all the workflows and applications in NGScloud2 using real datasets and have estimated the performance of the most cost-efficient instance for each process in the workflows. The datasets including the files produced by the validation tests are available at Zenodo repositories (DOIs: 10.5281/zenodo.4554359, 10.5281/zenodo.4554621, and 10.5281/zenodo.4554857).

Table 2 Tests of de novo RNA-seq workflow performance.

Output size and elapsed time (E.T.) consumed by each process and type of instance, number of CPUs and RAM employed to run each process. (*) Pcan1-S0_1.fastq.gz & Pcan1-S0_2.fastq.gz.

Process	OutputSize (MB)	E.T. (s)	Instance	vCPU	RAM(GiB)	
Uploading of compressed Pcan FASTQ files (*)	–	–	t3.medium (spot)	2	4	
Read quality assessment (FastQC)	2.09	500	
Cut of 12 nucleotides from the start of the read (Trimmomatic)	2586.08	1,474	
Trimmed read quality assessment (FastQC)	1.96	407	
Decompression of trimmed reads (gzip)	0.00	259	
Assembly of trimmed reads (SOAPdenovo-Trans)	3612.50	2,486	r5.2xlarge (spot)	8	64	
Assembly of trimmed reads (Trans-ABySS)	4019.22	40,727	r5.2xlarge (on-demand)	
Assembly of trimmed reads (Trinity)	60464.70	387,683	
Alignment of trimmed reads on the assembled transcriptome (Bowtie2)	3898.93	2,062	r5.2xlarge (spot)	
Quality assessment of the assembled transcriptome (BUSCO)	228.08	629	
Quality assessment of the assembled transcriptome (QUAST)	1.49	15	r5.xlarge (spot)	4	32	
Quality assessment of the assembled transcriptome using trimmed reads (rnaQUAST)	381.96	1,948	r5.2xlarge (spot)	8	64	
Quality assessment of theassembled transcriptome using trimmed reads (RSEM-EVAL)	128.07	1,979	
Quality assessment of the assembled transcriptome using trimmed reads (Transrate)	7291.58	2,036	r5.xlarge (spot)	4	32	
Assembled transcriptome filtering (CD-HIT-EST)	163.79	15,952	
Assembled transcriptome filtering (transcript-filter)	44.30	7	t3.medium (spot)	2	4	
Quantitation (eXpress)	76.10	320	r5.xlarge (spot)	4	32	
Quantitation (kallisto)	2476.32	1,168	
Variant calling (SAMtools & BEDtools & BCFtools & tabix)	1504.57	4,641	r5.2xlarge (spot)	8	64	

Data uploading, read quality assessment, trimming and pre-processing steps for RNAseq and RADseq workflows do not have excessive computational requirements (Tables 2, 3 and 4); therefore, they can be run efficiently using a cheap spot t3.medium instance (2 CPUs and 4 GiB of RAM). Other tasks, however, need the deployment of more powerful instances. For instance, a memory-oriented r5.2xlarge instance (8 CPUs and 64 GiB of RAM), has been used in the three transcriptome assembly processes tested in the de novo RNAseq assembly (Table 2).

Table 3 Tests of reference-based RNA-seq workflow performance.

Output size and elapsed time (E.T.) consumed by each process and type of instance, number of CPUs and RAM employed to run each process. (*) all files correspond to the 23 libraries of Pcan.

Process	Output size (MB)	E.T. (s)	Instance	vCPU	RAM(GiB)	
Uploading of compressed genome and annotation files of Pinus taeda	–	–	t3.medium (spot)	2	4	
Decompression of genome and annotation files	0.00	334	
Uploading of compressed Pcan FASTQ files (*)	–	–	
Read quality assessment (FastQC)	48.14	15,642	
Cut of 12 nucleotides from the start of the read (Trimmomatic)	67037.63	35,758	
Trimmed read quality assessment (FastQC)	45.21	11,631	
Decompression of trimmed reads (gzip)	0.00	9,169	
Trimmed reads alignment to Pinus taeda genome (HISAT2)	190994.33	73,516	r5.8xlarge (on-demand)	32	256	
Quantitation (htseq-count)	3.26	9,113	m5.2xlarge (spot)	8	32	
Variant calling (SAMtools & BEDtools & BCFtools & tabix)	52041.37	60,180	
Transcriptome alignment to Pinus taeda genome (GMAP)	3046.34	35,039	r5.8xlarge (on-demand)	32	256	

Table 4 Tests of RAD-seq workflow performance.

Output size and elapsed time (E.T.) consumed by each process and type of instance, number of CPUs and RAM employed to run each process. (*) SRR8199746_2.fastq.gz, SRR8199747_2.fastq.gz, SRR8199748_2.fastq.gz, SRR8199749_2.fastq.gz, SRR8199750_2.fastq.gz, SRR8199751_2.fastq.gz, SRR8199760_2.fastq.gz & SRR8199761_2.fastq.gz.

Process	Output size (MB)	E.T. (s)	Instance	vCPU	RAM(GiB)	
Uploading of compressed Q. ilex x Q. suber FASTQ files (*)	–	–	t3.medium (spot)	2	4	
Read quality assessment (FastQC)	8.11	357	
Cut of restriction site sequences (Trimmomatic)	1943.57	2,241	
Cut of Illumina TruSeq adapter sequences (cutadapt)	0.05	1,589	
Cut of overrepresented sequences (cutadapt)	0.05	1,441	
Cut of o poly(A) sequences (cutadapt)	0.05	1,321	
Read quality assessment (FastQC)	7.69	362	
Decompression of final trimmed reads (gzip)	0.00	100	
Pseudo-assembly of final trimmed reads (starcode) ->assembly 1	8402.62	13,874	r5.2xlarge (on-demand)	8	64	
Pseudo assembly of final trimmed reads (SOAPdenovo2) ->assembly 2	2010.87	1,907	
Final trimmed read alignment to assembly 1 (Bowtie2) ->alignment 1	20336.83	3,355	
Final trimmed read alignment to assembly 2 (Bowtie2) ->alignment 2	4217.00	1,702	
Final trimmed read alignment to assembly 1 (GSNAP) ->alignment 3	29994.83	954	
Final trimmed read alignment to assembly 2 (GSNAP) ->alignment 4	16676.63	1,672	
Variant calling using assembly 1 and alignment 1 (SAMtools & BEDtools & BCFtools & tabix)	7177.26	3,171	
Variant calling using assembly 1 and alignment 3 (SAMtools & BEDtools & BCFtools & tabix)	7458.21	3,152	
Variant calling using assembly 2 and alignment 2 (SAMtools & BEDtools & BCFtools & tabix)	840.43	532	
Variant calling using assembly 2 and alignment 4 (SAMtools & BEDtools & BCFtools & tabix)	7701.16	3,863	

For the reference-based RNAseq workflow, 23 pair-end libraries of an RNAseq experiment have been used in the tests, being the data volume much higher than in the de novo RNAseq workflow test. In this case, the most limiting step of the workflow consisted of read and transcriptome alignments performed with HISAT2 and GMAP, respectively (Table 3). These processes required an on-demand memory optimized r5.8xlarge instance (32 CPUs and 256 GiB of RAM), which is four times the instance employed for de novo RNAseq assemblies, to reduce the times needed to complete the alignments. For RADseq data, which consisted of eight small single end libraries, an on-demand memory oriented instance r5.2xlarge (8 CPUs and 64 GiB of RAM) was used in the read alignment with Bowtie2 and GSNAP (Table 4).

For functional annotation, TOA configuration and the external genomic database upload do not have large hardware requirements (Table 5). Therefore, a t3.medium instance type (2 CPUs and 4 GiB of RAM) was used, except for the database building processes of Refseq Plant proteome and NT and NR databases for BLAST+ and DIAMOND usage that have been run in a spot compute-oriented c5.xlarge instance (4 CPUs and 8 GiB of RAM) and a spot memory-oriented r5.xlarge (4CPUs and 32 GiB of RAM). However, attention must be paid to the provided storage when dimensioning the experiment, since the internal TOA database and intermediate files can reach >1.5 TB, mainly because of the size of the NCBI NR/NT databases.

Table 5 Tests of TOA configuration and genomic database processes performance in the functional annotation workflow.

Output size and elapsed time (E.T.) consumed by each process and type of instance, number of CPUs and RAM employed to run each process.

Process	Output size (MB)	E.T. (s)	Instance	vCPU	RAM(GiB)	
Creation of TOA config file	–	–	t3.medium (spot)	2	4	
Creation of TOA database	0.00	0	
Creation of genomic dataset file	–	–	
Creation of species file	–	–	
Download of other basic data	0.02	10	
Load of basic data into TOA database	0.02	57	
Build of Gymno PLAZA 1.0 proteome	0.02	73	
Download of Gymno PLAZA 1.0 functional annotations from PLAZA server	0.02	33	
Load of Gymno PLAZA 1.0 data into TOA database	0.02	281	
Build of Dicots PLAZA 4.0 proteome	0.02	280	
Download of Dicots PLAZA 4.0 functional annotations from PLAZA server	0.02	81	
Load of Dicots PLAZA 4.0 data into TOA database	0.02	1,100	
Build of Monocots PLAZA 4.0 proteome	0.02	234	
Download of Monocots PLAZA 4.0 functional annotations from PLAZA server	0.02	48	
Load of Monocots PLAZA 4.0 data into TOA database	0.02	404	
Build of NCBI RefSeq Plant proteome	0.02	286	c5.xlarge (spot)	4	8	
Build of NCBI BLAST NT database for BLAST+	0.02	4,843	
Build of NCBI BLAST NR database for BLAST+	0.02	7,900	
Build of NCBI BLAST NR database for DIAMOND	0.04	8 416	r5.xlarge (spot)	4	32	
Download of NCBI Gene functional annotations from NCBI server	0.02	7	t3.medium (spot)	2	4	
Load of NCBI Gene data into TOA database	0.02	1,532	
Download of InterPro functional annotations from InterPro server	0.02	2	
Load of InterPro data into TOA database	0.02	2	
Download of Gene Ontology functional annotations from Gene Ontology server	0.02	1	
Load of Gene Ontology data into TOA database	0.02	3	

TOA annotation pipelines that use BLAST+ or DIAMOND were run in a spot r5.xlarge (4 CPUs and 32 GiB) instance because the test data is a small dataset (Table 6), but larger instance types are required to make times shorter.

Table 6 Tests of TOA pipeline processes performance in the functional annotation workflow.

Output size and elapsed time (E.T.) consumed by each process and type of instance, number of CPUs and RAM employed to run each process.

Process	Output size (MB)	E.T. (s)	Instance	vCPU	RAM(GiB)	
Uploading of reference dataset file (MonolignolsGenes.fasta)	–	–	t3.medium (spot)	2	4	
TOA nucleotide pipeline using BLAST+ ->annotation 1	15.45	1,041	r5.xlarge (spot)	4	32	
TOA nucleotide pipeline using DIAMOND ->annotation 2	24.11	166	
TOA amino acid pipeline using BLAST+ ->annotation 3	32.13	4,997	
TOA amino acid pipeline using DIAMOND ->annotation 4	41.41	7,006	
Annotation merger of TOA pipelines using annotation 1 y annotation 3	5.49	7	t3.medium (spot)	2	4	
Annotation merger of TOA pipelines using annotation 2 y annotation 4	2.44	1	

Comparison with other cloud platforms for bioinformatics

Since Cyverse’s iPlant Atmosphere service has restricted access, and is currently only available to researchers based in the US, we have only performed a qualitative comparison of NGScloud2 and Galaxy (Table 7) regarding expenses, availability, hardware characteristics and other factors that may affect cost-efficiency. From the comparison between the characteristics of both platforms, it appears that they are complementary. Galaxy is an open access platform that allows using a wide set of bioinformatics software tools to define and run user-defined workflows for NGS data analysis (genome assembly, RNAseq, etc.) on small or medium volume datasets. By now, NGScloud2 has implemented de novo RNAseq, reference-based RNAseq, RADseq and functional annotation specific workflows for which the researcher can choose in a flexible way which bioinformatics application or applications to use in each task.

Table 7 Comparison between NGScloud2 and the Galaxy framework.

Galaxy employs several hardware infrastructure types (Galaxy cluster, Jetstream, Stampede2, PSC Bridges). Galaxy Data Source: https://galaxyproject.org/main/.

Feature	NGScloud2	Galaxy (https://usegalaxy.org/)	
Expense	According to use	Free	
Availability	Immediate	Galaxy cluster: short/moderade wait
Jetstream: short/moderade wait
Stampede2 (normal): moderate/long wait
Stampede2 (SKX): long/very long wait
PSC Bridges: moderate/long wait	
Maximum storage size (GB)	Unlimited	250 GB for registered users	
Data privacy	Strong	Moderate	
Maximum walltime (hours)	Unlimited	Galaxy cluster: 36
Jetstream: 36
Stampede2 (normal): 48
Stampede2 (SKX): 48
PSC Bridges: 24-96	
Maximum CPUs number	96	Galaxy cluster: 6
Jetstream: 10
Stampede2 (normal): 64
Stampede2 (SKX): 48
PSC Bridges: 5-20	
Maximum RAM size	768	Galaxy cluster: 30
Jetstream: 30
Stampede2 (normal): 96
Stampede2 (SKX): 192
PSC Bridges: 240-960	
Maximum concurrent jobs	Unlimited	6 for registered users	
de novo RNA workflow	Specific & flexible	User-designed	
Reference-based RNA-seq workflow	Specific & flexible	User-designed	
RAD-seq workflow	Specific & flexible	No	
Functional annotation workflow	Specific & flexible	User-designed	
Other workflows	No	User-designed	

One of the biggest disadvantages of Galaxy compared to NGScloud2 is its limited storage space (only 250 GB) that precludes running large experiments in an efficient way. Galaxy presents also limitations in the maximum number of CPUs and RAM that can be assigned to each process, and on the maximum number of concurrent jobs. Further, the user has to wait some time before the Galaxy infrastructure is set, leading to potential delays in getting the results of the analyses. Among NGScloud2 advantages, it is worth mentioning that AWS’s cloud storage is potentially infinite, and that hardware infrastructure deployment and termination are immediate, allowing running many instances in parallel. Therefore NGScloud2 can be used to securely analyze small, medium, large or very large data without waiting times or walltime limitations, always having in mind that it is a pay-per-use service.

Discussion

Bioinformatics is one of the fields that has benefitted from the development of cloud infrastructure. Many software applications contain instructions to be run on cloud systems (e.g., Eaton & Overcast, 2020 or https://github.com/bcgsc/transabyss/blob/master/TUTORIAL.md#16-mpi-and-multi-threading), but, very often, they require advanced knowledge about cloud infrastructure and how to configure it. Some other integral platforms are more similar to NGScloud2 philosophy, and aim to facilitate cloud instance deployment and data handling, at the same time they offer relatively easy access to software applications and workflow setting. In comparison, NGScloud2 presents some advantages, mostly related to the almost unlimited storage and strong hardware infrastructure availability, the immediacy of the hardware infrastructure setup, and the easy configuration of applications and workflows. Unlike NGScloud2, although Galaxy has a wide range of bioinformatics tools, it does not currently implement applications for the design of RADseq experiments or for the comprehensive analysis of these type of data. Moreover, the informatic system developed in NGScloud2 is flexible enough to allow for further incorporation of new software applications in the future.

NGScloud2 has proved its operability for RNAseq, RADseq and functional annotation analysis, but it is important to select the optimal application for each workflow step according to the specific type of data to analyze (Conesa et al., 2016; López de Heredia & Vázquez-Poletti, 2016) and the most cost-efficient instance to run each application (Mora-Márquez, Vázquez-Poletti & López de Heredia, 2018). Indeed, the instance type to use will depend on the hardware requirements of the bioinformatics software. NGScloud2 incorporates the possibility of using new generation r5, c5 and m5 instance types, that are more efficient and slightly cheaper than instances from previous generations. Once the instance CPU and RAM are chosen, the purchasing option of the instance (spot or on-demand) will depend on the time the process will be in execution, which pivots usually on the data size and on the algorithm complexity. It must be taken into account that the longer time an instance is running, the higher is the risk of being stopped. The current version of NGScloud2 allows restarting Trinity and SOAPdenovo-Trans processes from the point they stopped. However, most of the applications in NGScloud2 are not designed to be restarted after an unexpected interruption, a common issue in spot instances. Therefore we recommend using on-demand instances in processes with long elapsed times, while other processes that do not require long runtimes or can be re-started after interruption, such as variant-calling, are preferably run on spot instances.

For RNAseq and RADseq workflows, data uploading and read pre-processing are easily done with low-power instances, therefore at a minimal cost. Transcriptome assembly and mapping to reference genome are the most limiting task of the workflows (Miller, Koren & Sutton, 2010; Mora-Márquez et al., 2020), both in terms of the storage required for intermediate and output files and of the RAM and CPU number required to run the application. In these cases, memory oriented instances are more efficient than compute-oriented instances because the intrinsic nature of the assembly algorithms. It must be stressed out that memory and CPU requirements of de novo RNAseq assemblies grow linearly with read number for Trinity, and exponentially for SOAPdenovo-Trans (Mora-Márquez et al., 2020).

For functional annotation with TOA, the main hardware limitations are produced in the BLAST+ or DIAMOND homology search step that requires a minimum of 4 CPUs and 32 GiB instances for small datasets, but larger instance types are required for bigger datasets (e.g., a full transcriptome) or to reduce runtimes. It should be also noted that DIAMOND pipelines can be run in shorter times than BLAST+ pipelines (Buchfink, Xie & Huson, 2015; Mora-Márquez et al., 2021). A sufficient provision of storage volumes is also recommended to run the functional annotation workflow, to take full advantage of TOA capabilities.

Conclusions

NGScloud2 has significantly expanded the types of bioinformatics workflows to run using Amazon Web Services since its previous version. This new version has incorporated major technical improvements that optimize the use of popular software applications otherwise inaccessible to non-specialized users lacking suitable hardware infrastructures. Moreover, these technical improvements are oriented to significantly reduce costs by simplifying data access and taking advantage of EC2 spot instances that may produce savings of up to 50–80% in many steps of the analysis. Therefore, NGScloud2 constitutes a good alternative to other cloud-based platform to analyze RNAseq and RADseq data in model and non-model species.

We would like to thank M. Hurtado and Dr. V.M. Chano for beta-testing of NGScloud2 operability.

Additional Information and Declarations

Competing Interests

Author Contributions

DNA Deposition

Data Availability

The authors declare there are no competing interests.

Fernando Mora-Márquez and Unai López de Heredia conceived and designed the experiments, performed the experiments, analyzed the data, prepared figures and/or tables, authored or reviewed drafts of the paper, and approved the final draft.

José Luis Vázquez-Poletti conceived and designed the experiments, analyzed the data, authored or reviewed drafts of the paper, and approved the final draft.

The following information was supplied regarding the deposition of DNA sequences:

Read data are available at NCBI SRA: SRX5228139–SRX5228161 for Pcan, and SRX5019123–SRX5019138 for Suberintro.

The following information was supplied regarding data availability:

NGScloud2 code, instructions for software installation and use are available at https://github.com/GGFHF/NGScloud2;

NGShelper is available at https://github.com/GGFHF/NGShelper.

Illumina Read data are available at NCBI SRA: SRX5228139 -SRX5228161 for Pcan, and SRX5019123-SRX5019138 for Suberintro.

Datasets including the files produced by validation tests are available at Zenodo:

Mora-Márquez, F., Vázquez-Polletti, JL., & López de Heredia Larrea, U. (2021). NGScloud2 - Validation tests 1: de novo RNA-seq, RAD-seq, functional annotation [Data set]. Zenodo. http://doi.org/10.5281/zenodo.4554359

Mora-Márquez, F., Vázquez-Polletti, JL., & López de Heredia Larrea, U. (2021). NGScloud2 - Validation tests 2: reference-based RNA-seq (part 1) [Data set]. Zenodo. http://doi.org/10.5281/zenodo.4554621

Mora-Márquez, F., Vázquez-Polletti, JL., & López de Heredia Larrea, U. (2021). NGScloud2 - Validation tests 3: reference-based RNA-seq (part 2) [Data set]. Zenodo. http://doi.org/10.5281/zenodo.4554857.

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
