# Peer review of "NGScloud2: optimized bioinformatic analysis using Amazon Web Services"

_PeerJ, doi:10.7717/peerj.11237_

## Round 0.1 · original submission · Major Revisions

We have received three positive reviews on this work demanding some updates. The reviewers suggested a more detailed comparison of the available tools. I recommend restructuring the Results section and comparing to recently published tools as suggested by Reviewer #3.

Only PeerJ publishes Bioinformatics Tools so it is not appropriate for PeerJ Computer Science.

Waiting for resubmission of the revised manuscript.

Reviewer 1 ·

Basic reporting

Language throughout is clear, professional, and unambiguous, though I highlight some typographical errors in "General comments".

Intro and background insufficiently show context for the work. For example, much more could be done to elaborate on "Next Generation Sequencing (NGS) technologies and the algorithms and applications employed". Overall the Introduction feels very rushed.

The results and discussion appear to be largely methods, in my view. I would move all the content of results and discussion to methods, and then provide more results and discussion as relevant.

Much could be done to improve accessibility of this framework by providing more information for the reader about the details of AWS cloud infrastructure. For example, on line 92 "architecture of instances and volumes" will be entirely opaque to the average reader. What is a "volume"? What is an "instance"? What are "m4, c4 and r4 instance types"? Try to clarify technical jargon for the non-specialist.

The figures are very complicated, do not have legends, and are not described sufficiently in the main text, so it's difficult to understand their meaning.

Raw data is supplied in the SRA, which I verified.

Experimental design

The correct operability of the pipeline is referenced in the materials and methods, yet no evaluation of this is presented in the results. I would like to see more elaboration on the methods for this evaluation and presentation of the results.

Typically a new method will involve some validation with simulated data to ensure correct results. I would recommend this here.

Validity of the findings

Results and discussion are largely methods, so I can't evaluate validity of the findings.

Additional comments

The authors present NGSCloud2, a bioinformatic system for interfacing with Amazon Web Services (AWS) to facilitate running next-generation sequencing analysis workflows on flexible AWS cloud computing infrastructure. It is clear that the authors have done a significant amount of work to provide this resource to the community, and I applaud their effort. The documentation is extensive, which is wonderful, but the install process may be beyond the reach of non-specialized users, so I would recommend trying to ease this burden. As it was not part of the manuscript, I did not consider the state of the documentation or the install process during my review, I only highlight this here for the benefit of the authors.

L53: I would not call AWS a "public cloud service" as this is a private service which the user must pay for.
L60: M3, C3, and R3 will not mean anything to the average reader. Can you make it more clear what the difference is between these?
L65: Delete "NGSCloud2" here as it's redundant.
L75: As the methods for these bioinformatic pipelines haven't been described yet, it's a little confusing to be referred to the Results and Discussion on this line.
L93: "Figure 1" <- Add space

Table 1
"Read alignment" <- Bowtie2
"Transcriptome filtering" <- CD-HIT-EST

Table S1
I know supplementary materials are not supposed to be reviewed, but this appears to be a pricing list for AWS cloud compute usage. I don't think this is a result of the work of the authors, and I also think this will become rapidly outdated, so I would suggest removing.

Reviewer 2 ·

Basic reporting

This work presents next-gen sequencing analysis tool based on Amazon Web Services (AWS) platform. The work is clearly presented. English is Ok.
Raw data (SRA) are shared. The manuscript has sufficient (but rather short) background for cloud computing and existing tool. The introduction could be extended.
Some abbreviation might be not understandable for broad readers’ audience (“m4, c4 and r4 instance...”)
The work meets PeerJ standards, overall.
However, reader can see just presentation of the tool, but not the novel results. It might be appropriate for bioinformatics paper. I believe PeerJ is the journal for Life sciences research rather than just software tool description.
PeerJ Computer Journal seems more appropriate for such publication (Just suggestion to consider)

Experimental design

The work describes the pipeline, but not biological results. I expect more biological applications rather than just software updates in the Results. Presentation of the Methods section is quite detailed.

Validity of the findings

This work presents NGSCloud2, the bioinformatics tool for cloud computing. As the software it is quite useful, presents novel solution and diversity current platforms in sequencing analysis. Thus, such work is valid for science community and should be published.
However, the findings/results are not given in classical form. IT makes this paper interesting for the specialists, but not a reader with general background.
Computer jargons and unclear technical presentation including figures should be fixed.

Amazon Web Services named as "public cloud service" though it is commercial project. Please state it clearly. Supplementary Table S1 with the pricing looks not relevant for science publication. Need either remove or update it.
Please comment on the abbreviations M3, C3, and R3.
And explain CD-HIT-EST.

Additional comments

Please avoid technical details to make this work close to broad reader audience. The Results section should be reformatted.
The figures' legends could be more informative.

·

Basic reporting

no comment

Experimental design

no comment

Validity of the findings

no comment

Additional comments

Fernando et al provide an updated toolkit to analyze RNA-seq data in Amazon cloud. The toolkit adopts several most popular packages for RNA-seq analysis and the author shows a distinct flowchart and structure on Amazon platform. The online resource and service faciliate the personal bioinformatic analysis and teamwork in different loci. I agree the authors provide a solution of simple and convenient protocols for RNA-seq data and the manual is quite readability for a beginner on Amazon cloud.
However my concern is about the usage or experience for the user. Currently, we have too many choices for the tools selection (https://en.wikipedia.org/wiki/List_of_RNA-Seq_bioinformatics_tools) and hardly find a best pipeline for all types of data(Genome Biology.2016,17:13). Some tools have limited maintenance and are out-of-date in scientific community (eg. "cufflinks" for DEG calling). There are still some popular tools for DEGs calling (Bioinformatics.2019,35(18):3372–3377 ). I suggest author to list the activity status for database (eg: https://bigd.big.ac.cn/databasecommons/) and the latest version info to provide additional information for the user. Another suggestion is to add the steps of "enlarge the volume in EBS" in the manual book.

---

## Round 0.2 · accepted · Accept

The reviewers have no more critical remarks on the text. However, there are some comments on text organization (except typos). I believe the authors will fix it in the final version. I endorse the publication of this manuscript.

Reviewer 1 ·

Basic reporting

no comment

Experimental design

no comment

Validity of the findings

In my view, the section "Comparison with other cloud platforms for bioinformatics" isn't really a "result". I would move this to the discussion.

Additional comments

The authors have satisfactorily addressed all my initial comments and suggestions. The added text describing NGS and AWS in the introduction are very helpful. My main remaining concern is the redundancy between the results and the discussion. For example, the first paragraph of the discussion
largely revisits the "Comparison with other cloud platforms for bioinformatics", and the last two paragraphs of the discussion largely recapitulate the results section "Validation of NGScloud2 operability". I suggest reorganizing the results and the discussion to reduce the redundance.

Line 264: "functional"

Reviewer 2 ·

Basic reporting

Thank you for the manuscript update. It is interesting material from the point of view of a user.

Experimental design

Experimental design of the application fits to the journal scope.

Validity of the findings

Novel software tool is presented, useful for wide range of classical NGS applications in transcriptomics. The results are based on the data, well reproduced. All the tests and resulting data are provided in archive, stored in Github.

Additional comments

Due to extensive text update there are some logic duplication (repeating phrases in the discussion part of the text (advantages of NGScloud2). It gives no new information. Recommend remove it or rephrase.

There are some technical comments to be fixed before publication.
RNA-seq name usually is published with a dash. But the authors use RNAseq throughout the text, except of Table 1. I recommend at least write first in the text “RNA-sequencing (RNAseq)” to keep consistency.
I have no more critical remarks. Endorse publication of this manuscript.
Line 29: de novo - could be in Italic
Line 81: “several applications” - please mention several, not only Galaxy. Or remove word ‘several’.
I’d recommend mention iPlant, miCloud (doi: 10.1089/cmb.2018.0218), DolphinNext (doi: 10.1186/s12864-020-6714-x) to name a few.
Line 100: GiB - it is GiB through the text, more common is GB abbreviation. May keep as is, however
Line 111-112: Mora-Márquez, Vázquez-Poletti & López de Heredia U, 2018
The citation could be by first author name only, with ‘et al.’
Line 131 - repeat of the full abbreviation from line 121 -
Restriction site Associated DNA sequencing (RADseq)
- may just use ‘RADseq’ second time
Line 132: and functional annotation -
More common is ‘gene functional annotation’
Line 142: ‘n virtual machines’ - need mark ‘n’ in Italic.
Maybe specify here maximal (tested) number of n machines. Or remove it, since number n is not used anymore in the text.
Line 180: JCR -
Please add the link (https://www.scimagojr.com), or full abbreviation for JCR
‘Web-of-Science’ is written with dash, or as ‘WoS’. Might give here reference or link to the database used.
Line 212: (Anders, Pyl & Huber, 2015) - citation could be by first author name et al.
Line 231: psedudogenome - typo in ‘pseudo’
Line 240: TOA - give the abbreviation in full (it is given in the Table only)
Line 241: (PLAZA) - the database could be with the link or reference
Line 242: NCBI's RefSeq
May write NCBI without ‘s’
Line 251: Gene Ontology -
There are different ontology databases, please give the link (http://geneontology.org/ or preferably direct to the data source)
Line 265: Illumina TM -
I think it is quite standard, may write without (TM), or at least mention only once, not need repeat in line 267.
Line 292: PE -
Please comment as (pair-end) since there are two many abbreviations in the text, it is better avoid extra abbreviations.
Same for line
298: SE
- write single end, minimize the use of abbreviations
Line 305: compute oriented
- compute-oriented
Lines 307-308: ‘because’ is repeated, try to rephrase
‘TOA.db’ maybe write as TOA database? This name is not used as .db in the text
Line 322: ‘in a flexible way’ - what means flexible? Could give more detail here since it is discussed as the advantage of the tool.
Maybe ‘flexible’ in terms of user time? Or optimal in some other sense?
Lines 346-350: --
I believe it is redundant text repeat, already mentioned in the text section before.
Rephrase or just remove it for clarity.
Line 417: ‘Table 1’ - it is not paper title, check the reference
There is some improper reference formatting for line 454 too.
Figure 5 -
A plot right to ‘Summarize & plot’ box - it is too small, not clear what is it? Assume it is just illustration of a plot?

Overall all the remarks are really minor for authors’ discretion to fix.
I will not check it the text again.